# The Politician: Action and Creation in the Practical Ontology of Gilles Deleuze

Julian Ferreyra

National Council of Scientific and Technological Research, University of Buenos Aires, Caba C1053, Argentina; djulianferreyra@gmail.com

**Abstract:** This paper addresses an action that, from a Deleuzian perspective, is capable of modifying the despairing current social situation in which we are immersed, through the creation of political Ideas. Even though Deleuze conceives social Ideas as vast civilizing structures, we propose to bring into the political domain the logic of other acts of creation, such as the artistic or the philosophical, where the monumental coexists with minor figures that are nonetheless capable of introducing novelty into the world. The politician is the figure of those who are capable of having an Idea that allows to break the habits that perpetuate the current situation, and gives consistency to the intensive forms of life that continually create and dissolve themselves in the flow of becoming. Thus, macro- and micro-politics do not oppose each other, but offer in their immanence an alternative to social nihilism.

**Keywords:** Deleuze; political ideas; action-image; intensity; macro-politics; micro-politics

## 1. Introduction

"Doubtless, the present situation is highly discouraging", Deleuze and Guattari wrote in 1980 [1] (p. 422). Forty years later, their diagnosis has not improved: the COVID-19 pandemic is the straw that broke the back of the camel of denial. Our social situation forms a "pathogenic milieu" where the community can no longer develop any "illusions about itself" [2] (p. 147). Deleuze and Guattari have tried to think of this milieu, following the Marxist tradition, using the concept of "capitalism". The problem is that capitalism is part of a series of large social organizations, forms of *socius* or Apparatus of Capture, and therefore is something too large for us: it is easier to imagine the end of the world than it is to imagine the end of capitalism (according to the phrase that Mark Fisher attributes to both Frederic Jameson and Slalov Zizek [3] (p. 2). The result is mounting nihilism: "one persuades oneself that one has no choice" [2] (p. 114), and that the only thing we can do is adapt to this hostile socio-ecological milieu. This adaptation, however, implies accepting all the consequences of capitalism in its latest form, the "capitalocene" [4] (p. 6): social inequality, looming environmental catastrophe, the spread of disinformation and the ensuing paranoia, loss of sense and value and each and every one of the deaths that the policies of "economy before health" have caused in the context of the COVID-19 pandemic. Cynicism and defeatism take over the world: "Both the Anthropocene and the Capitalocene lend themselves too readily to cynicism, defeatism, and self-certain and self-fulfilling predictions, like the 'game over, too late' discourse I hear all around me these days" [5] (p. 59).

In the face of this situation, the practical-political question that emerges is a classic one: What is to be done? How can we get rid of defeatism and the feeling of inexorability? How do we get rid of the common sense that makes it easier to imagine the end of the world than the end of capitalism? In these pages, we will delve into the philosophy of Gilles Deleuze for the tools to get some answers regarding political action. While in many other conceptual frameworks (as for example Hanna Arendt's), action as a way of bringing something new into the world, and thus transforming the situation, is a truism, the topic poses a number of conceptual conundrums for the philosophy of Gilles Deleuze. However, tackling the issue

can allow bringing a new perspective. In order to achieve this, we will track the notion of action in Deleuze's work, mainly by focusing on the concept of "action-image" that Deleuze expounds in his studies on cinema (his most extensive treatment on the topic of action). This concept enables a passage from the present situation (S) to a transformed situation (S′), as expressed in the formula S-A-S′ (to which we will return in the pages that follow). From this point of view, the "large form" is much more than a category in a classification of the history of cinema: it becomes a political alternative that disrupts the contraction characteristic of the hegemonic habit. Considering that, as we will show, the action-image in *The Movement-Image* resonates strongly with the first synthesis of time in *Difference and Repetition*, it is legitimate to relate the elaborations on cinema with the fundamental ontology developed in the 1968 book. The first synthesis takes place in a particular dimension of the Deleuzian ontology, the extensive field; but we will show that, in order to think an action that is able to actually transform a situation, it is mandatory to articulate it with the other ontological dimensions: intensity and Ideas (extension, intensity, and Ideas being the three main fields covered in *Difference and Repetition*, as I have developed in detail elsewhere and will cover briefly below) [6]. Dale Clisby gives a thorough account of the main positions in Deleuzian studies regarding the different fields of his ontology and their relation [7]; in this regard, the positions of Smith [8] (pp. 59–85) and Williams [9] (pp. 138–164) provide useful resources. However, the most accurate interpretations are those of Santaya [10] and Mc Namara [11], the latter clearly showing that, against all the "standard" literature, intensities are actual (p. 58). Only by keeping in mind the immanent articulation of the three ontological fields can an action intervene in reality and render a true act of creation possible.

In order to embrace the complexity of this issue, we will resort to Deleuzianism in Latin America. Most of its outstanding figures tend to empathize with only one dimension of political ontology: the intensive, whose corresponding question would be how to transform our way of life through the creation of affections and micro-political actions. For this line of thought (that can also be found in Deleuzians all over the world), the question is "*changing life*", as Arthur Rimbaud pointed out [12] (p. 281). Doubtless, there is action as well as creation in this dimension (which is likely the richer and most prolific in the work of the French philosopher). However, the intensive way, by itself, is bound to resistance, guerrilla, and a life in the fissures and the blind spots of a situation that is still highly discouraging, that does not transform itself, and that—as we will show—once and again leads us to the worst automatisms. Capitalism, meanwhile, is "apparently victorious" [13] (p. 139) by a combination of macro- and micro-politics that crushes, corners, and sickens the forms of life that are created through micro-political actions. In this "intensive" perspective, the philosophy of Gilles Deleuze appears to be more useful to place ourselves in a position where the political and ecological global catastrophe can lose its weight (the comforting predictions: at least we were right, the world was coming to its end), than to offer an option for transforming reality through politics.

Indeed, if one suppresses in oneself "everything that prevents us from slipping between things" [1] (p. 280), is there any importance left in the shame of being a man, of the roar of factories and bombings, of the sanitary and environmental catastrophe? Our planet is only a rock among other rocks, and life is a minor phenomenon in the great chain of beatitude. Here, we will explore a different perspective, which focuses on the usefulness of action and its ability to transform reality, following the definition of political philosophy offered by the Argentinean thinker Damian Selci in his *Theory of militancy*: "if there was a thermometer for political philosophy, it should measure its usefulness for popular struggles ... The political results *are* the capacity for the transformation of reality" [14] (p. 21). Rimbaud's phrase must be completed, in the very same way the surrealists did in the early 20th century: "'Transform the world', Marx said; 'change life', Rimbaud said. These two watchwords are one for us" [15] (p. 241). It is not enough to change life (intensively); it is also necessary to change the world (extensively), that is, to transform reality in such a way that the production of ways of life is no longer evanescent. To say it the other way around: without the perspective of new and desirable ways of life, why transform the world?

However, the present situation is, once again, highly discouraging. One is *forced* by the "physical necessity (the state of things, the situation)" [2] (p. 114). One actually has no choice. We have to adapt to capitalism, acquire its habits, and submerge in its continuum in order not to perish. Yet, according to Deleuze, this "there is no choice" position points to the fundamental choice between modes of existence: the "choice of choice or non-choice" [2] (p. 114), but how do we "choose" when the situation is highly discouraging, suffocating, and pathological (where the community can no longer develop any "illusions about itself")? We choose, precisely, through creation; and when the question is to transform reality, a *political* creation becomes necessary. As we will see later in this article, political creation, in the very same way as artistic and philosophical creation, demands "having an idea" [16] (p. 312).

What does it mean to have an Idea in politics? In his political writings (notably, the brief section about the social Idea in *Difference and Repetition* [17] (pp. 186–187), the third chapter of *The Anti-Oedipus* [13] (pp. 139–271) and the chapters on the War Machine and the Apparatus of Capture in *A Thousand Plateaus* [1] (pp. 351–473)), Deleuze presents a quite narrow image by restricting political Ideas to the macro-political dimension, the great structures: capitalism and its axiomatic, the State and its heads, the Despot and its bureaucrats, etc. (we will show how Deleuzian writings support this affirmation of the macro-political trait of the social Ideas in Section 5). By doing so, he only increases discouragement, capturing us between a situation where there is nothing we can do (nihilism) and the moral imperative of a revolution that is both impossible and sacrificial. That is the criticism that Selci aims at Deleuze: "The enemy was so big, and the character of the emancipatory struggle so 'local' and small, that philosophy seemed to be content with supplying a sophisticated theory for lucid resignation" [14] (p. 11). According to Selci, Deleuze and all the "post-structuralists" are *theoretically* fouling the air of political action.

The key to free ourselves from that trap is *a difference*: Deleuze restricts Ideas to the great social structures only in the *political* field. In the rest of the creative fields, "to have an idea" points in another direction, towards creations that are more modest and minor, but nonetheless capable of actually modifying the situation. By making a bridge between the ontological Idea of *Difference and Repetition* and the creative ideas of "What is a creative Act?", we will try to show that, in the very same way that the artistic and philosophical creative acts produce movements, lineages, and schools, the political creative act can produce, within the situation, lines of political action that are capable of modifying it. Such is the necessary optimism to go into politics. It is a matter of exploring the field of political Ideas, of thinking about the history of "those who have that Idea", that is, their signatures. These are the politicians: the men and women who have Ideas, bring them into the world, and change, for better or worse, the concrete lives of the human beings that populate it. Politicians carry their extensive actions on, not in an isolated way, but as a part of a larger whole. That is the spirit of the figure of the "militant" in Selci, that is the alliance between macro- and micro-politics in Rolnik, and (as we will see) that is the role of the State in the possibility of the "wild life" as resistance to the "*vida mula*" proposed by the collective of thought Juguetes Perdidos. This will require introducing a particular interpretation of both the ontological and the creative Idea that is not always in line with the established secondary literature.

In this path, it is necessary to free the political philosophy of Deleuze from the double bind of macro- and micro-politics. There is an axiological trait in those who remain trapped in that double bind: "good" micro-politics against "diabolic" macro-politics. To dismantle that double bind and show that they are immanent dimensions (as we will see in our Conclusions) is the only way to make Deleuze offer a genuine contribution to the political dilemmas that become more and more pressing. It is not so much a matter of taking sides, as it is of understanding that the immanence of both dimensions of the political is the only way to accomplish the motto of the surrealist revolution: transform the world *and* change life. For us, micro-politics and macro-politics, intensities and extensions, flights and individuations, make one and the same political stance to break out of the highly discouraging present situation.

## 2. Extensive Action: The Large Form and the "*vida mula*"

The notion of "action" receives a restricted treatment in the work of Deleuze. In his early *Nietzsche and Philosophy*, action appears briefly, subordinated to the affirmation and the active-becoming that it implies (with the possibility of being also a mere instrument of nihilism [18] (p. 54)); in the *Logic of Sense*, we also find "action" in a subordinate position, this time in regard to the event [19] (pp. 207, 245) or to the "noematic attribute" (p. 221). In *Difference and Repetition*, action finds its place in the first synthesis of time, that of the present and the habit, as being constituted by the contraction of the elements of repetition [17] (p. 75); but also in the "tremendous" action that is "too big for me" and leads to the third synthesis of time (p. 89). In *A Thousand Plateaus*, action appears as the possibility of constructing a rhizome as "political action" [1] (p. 12) and poses the concept of "free action" that, as opposed to "work", opens up emancipatory possibilities (pp. 397–398). In short, action is either subordinated to an a-subjective, intensive field of pure differences or becomings, or it is placed *directly* in that field. In all cases, this notion receives a cursory treatment and does not occupy a central position.

The most lengthy and detailed treatment of the notion of action appears in the first volume of the books on cinema, *The Movement-Image*. In this volume, one of the three varieties of movement-images that Deleuze discriminates is the action-image. It is the "large form" (expressed by the formula SAS'): "from the situation to the transformed situation via the intermediary of the action" [2] (p. 142). The situation is "a set of power-qualities as actualized in a milieu, in a state of things or a determinate space-time" [2] (p. 142). The situation is bound to the extensive field in the ontology of *Difference and Repetition*, for it is characterized by actualization in determinate spaces and times: discrete, measurable, closed-into-themselves multiplicities connect with other extensive "parts" in an extrinsic way; they divide without changing their nature; they can be measured, ordered, and segmented [17] (p. 223).

The extensive is the privileged field for *political* action. Indeed, while other forms of action (for example, aesthetic action) can reach their climax in the virtual or intensive fields (leading us towards the intensities imprisoned in our bodies), there is no political action that can do without extensions. The stomachs that growl with hunger, the skins that are flayed at the mercy of the elements, the sicknesses that break out without sanitary networks to contain them, the violence that blows up without shelter for the weak…They are all focuses of intervention for political action. Intensities do not nourish, they do not heal, do not shelter.

This raises the following theoretical issue. As we will see, according to Deleuze, the *extensive* field implies degradation and entropy. Nonetheless, whoever *acts* in this field cannot hang their head and give up ("it's too sad") or just wait for a lucky strike that can locally decrease entropy (faith and love). In the action-image, the agents actually act according to their evaluation of the situation, in order to *transform* it. They turn the situation into the origin of a new situation. They depart from a determinate situation, some kind of action emerges, and a modified situation takes place: SAS'. Deleuze details the film genres where this is predominant and distinguishes the various points of departure and forms of action, and even the possibility that the situation *does not* transform itself (SAS). The question is how this is possible, how an *effective* (and therefore transformative) action can take place in an extensive situation that is burdened with habits (which drive it to endure) and entropy (which forces it to degrade).

The tension between the transformative action and its effective possibility is immense. Even considered in a realistic and empirical manner, there is no *subjectum* of the action, that is, the action is not grounded in a Subject who acts, and who would be its principle or *ex-nihilo* foundation. The action is not "like a shot from a pistol" [20] (p. 45), but includes a People or a "makeshift group", and also "the objects adjacent to the situation", among other elements [2] (pp. 154, 158). The *situation* of origin also modifies the possibility of acting upon it; something in the very determination of the situation makes it possible for the individual who *acts* to emerge (or, on the contrary, inhibits them, making the action

impossible). This means, precisely, an obstacle for the action itself. Indeed, it is not always possible to find how to "respond to the challenges of the milieu as to the difficulties of a situation" [2] (p. 144); it is not always possible to perform an action capable of transforming the situation, because "all the *milieux* are pathological, and all modes of behavior are cracked" [2] (p. 145). In fact, the idea of a person who knows how to respond to the situation is nothing more than a *dream* (the "American Dream"). In truth, the situation leads the game. The *authentic* form is SAS. The only outcome of the action is the same situation that preceded it: "the hero becomes equal to the milieu via the intermediary of the community, and does not modify the milieu, but re-establishes cyclic order in it" [2] (p. 146)—or, in any case, the action leads the situation to its degradation, to an S' that becomes increasingly discouraging (S'', S''', until the heat death of the universe, the end of the world, the apocalypse).

After all, as we saw, the situation as "a set of power-qualities as actualized in a milieu, in a state of things or a determinate space-time" [2] (p. 142) points towards Deleuzian ontology's extensive field. "Milieu", "state of things", and "determinate space-time" connect with the extensive aspect of actualization, in terms of *Difference and Repetition*. The *milieu* is a *relatively* closed system, where a small set of factors are selected, specifically a determinate space-time: what is experienced in *that* closed *milieu* is a "state of things". There, a phenomenon of exception, of counter-weight against the fundamental ontological law (that is, according to Deleuze, *difference*) is produced. In the "state of things" that is produced in the closed milieu, the constitutive differences tend towards identity, they degrade themselves following entropy, which goes from the most differenciated to the less differenciated: "intensity defines an objective sense for a series of irreversible states which pass, like an 'arrow of time', from more to less differenciated, from a productive to a reduced difference, and ultimately to a canceled difference" [17] (p. 223). Thus, the generalities and the laws of nature emerge. Thus, individuals and genres emerge. Thus, the perseverations and the constants emerge. "No doubt there are as many constants as variables among the terms designated by laws, and as many permanences and perseverations as there are fluxes and variations in nature" [17] (p. 2). In brief, the different aspects of the extensive field such as it is presented in *Difference and Repetition* (whose immanence with the intensive and virtual fields will be dealt with below) are taken up again in the characterization of the *situation* as the first moment of the action-image.

Now, following this train of thought, which is the mode of *action* that is possible? Which action in *Difference and Repetition* is bound to the permanences and perseverations, to the determinate times and the relatively stable spaces? Which is the action that transforms repetitions into identities? It is *habit* [17] (p. 225) as *passive* syntheses "which render possible both the action and the active subject" [17] (p. 75). What makes action possible is our habit of living, our expectation that "this" will continue, the perpetuation of "our" case: "Passive synthesis is of the latter kind: it constitutes our habit of living, our expectation that 'it' will continue, that one of the two elements will appear after the other, thereby assuring the perpetuation of our case" [17] (p. 74). However, in this context, the possibility of a transformative action does not seem very plausible. On the contrary, the *milieu* will preferentially determine the actions that tend to perpetuate our case and to prolong at all costs the constructions that we were able to build upon the ontological precariousness.

This has, without question, a positive and even necessary side. What kind of politics could not care about the task of assuring the actually existent lives? However, in the present *situation*, the prevailing effect of governmental politics is the perpetuation of modes of existence that bring only suffering and misery. A most precise theory of the new habits and their sense in the discouraging current situation is offered by the Argentinean collective of thought Juguetes Perdidos, formed by Leandro Barttolotta, Gonzalo Sarrais Alier, and Ignacio Gago. This collective offers a Deleuzian perspective on a reality that Deleuze himself could only perceive intuitively at the time of his death, more than two and a half decades ago in the French context, which is geographically, socially, and historically very remote from the context of the most impoverished neighborhoods in the southern extreme of South

America. To think about the chain of habits through which "our" case is perpetuated in the conditions of the current phase of capitalism, they create the concept of "*vida mula*" (the life of a mule). Let's try to translate a passage that is full of Argentinean argot:

> [*Vida mula*] is a continuum of work, consumption, family, education, social aids, and a lot of daily businesses that are conducted to stay afloat on the precariousness of each of those vital aspects . . . That is what the one that *mules* does; he carries the load of the precarious job, but also of family troubles, the need for consumption, the violence of the neighborhood, the social discredit, the gratuitous corporal unease, the crammed commuting in trains and buses. [21] (p. 16)

It is the life of a mule: we carry existence much like Nietzsche's ass, without considering the creative horizon that is an essential part of our residence on earth.

> *Vida mula* as a continuum of work, consumption, request for quietness, family, neighborhood reality, moral codes, Pope Francis*m*, couple life, etc. links images, scenes, lives . . . *Vida mula* is a relentless chain of work (more or less precarious depending on each case), consumption, numb life, the emptiness at one's back, fragile stabilities and subjective closures (how to rest in the role of citizen, of neighbor, of good boy or bad boy, of worker, and stay there) . . . *Vida mula* as shackles, then, that aim to close an indeterminate, tiring sky and to ward off a precarious ground made of frightened and nervous moods, wearisome daily businesses (commute, jobs, housing, relations) that amount to very little. . . . [22] (pp. 56, 83–83)

*Vida mula* as described by Juguetes Perdidos implies no moral judgment. They do not refute it in the name of a transcendent ideal; on the contrary, they present its links as quite reasonable actions, taking account of the conditions of contemporary life, its constitutive precariousness, especially in the most vulnerable neighborhoods that surround the city of Buenos Aires in Argentina. The vital alternative is as follows: "What is best, a shelter in the *vida mula* or the exposure to the infinite and to precariousness?" [22] (p. 84). In the face of this alternative, *vida mula* is a healthier option than the exposure to precariousness and emptiness. It is a *bad life* (*mala vida*), but it is the life that is possible in this situation: "*Vida mula* is the Reality" [21] (p. 51). What is pathogenic is the milieu, the situation, as Deleuze points out:

> This would be the great difference between healthy and pathogenic milieux. Jack London wrote fine passages in order to show that, finally, the alcoholic community has no illusions about itself. Far from producing dreams, alcohol "refuses to let the dreamer dream", it acts as a "pure reason" which convinces us that life is a masquerade, the community a jungle, life a despair (hence the sneering of the alcoholic). The same could be said of criminal communities. On the contrary, a community is healthy in so far as a kind of consensus reigns, a consensus which allows it to develop illusions about itself, about its motives, about its desires and its cupidity, about its values and its ideals: "vital" illusions, realist illusions which are more true than pure truth. [2] (pp. 147–148)

The spiral closes onto itself: only wrong actions in a pathogenic situation. The actions perpetuate the situation, while the situation demands the continuity of such actions. To change the world, it is necessary to have illusions about ourselves as a society. We need optimism. Following a famous saying of Argentinean former president Cristina Fernández de Kirchner during an interview, Damian Selci focuses on the bond between this need for optimism and political action:

> [As militants] *optimism is an obligation* . . . It is not a consequence of an analysis of reality, on the contrary, it is its cause: precisely because I am an optimist, I analyze reality according to my optimism, and this is possible only because I include a factor that is invisible for the neutral-intellectual regard: *my own intervention in reality*. [14] (p. 118)

The concept of "militant" that organizes Selci's theory is optimist, as far as it points to an agent that is actually capable of an action A that leads from the situation S to the transformed situation S′ and, by achieving this, breaks with the depressive-deathly line that lets itself be crushed under capitalist realism [14] (p. 114). It aims to break free from the *vida mula*. And yet, where does this capacity for breaking free from the life of a mule come from? According to Selci, this capacity stems from a *change of life*: a complete politicization that slashes through the *continuum*: "[Organization] asks the militant to achieve a complete politicization . . . He must resign to the richness of his existence: if he was a worker, professor, son, mother, musician, scientist, lazy or hardworking, whatever his religion, with his gender or ethnicity, character, mood, it must all be sacrificed for the Grey Universal of politics" [14] (p. 130). This is Selci's blind spot: sacrifice is not enough, for it never gives way to the illusions that a *healthy* community must have about itself in order to make a transformation possible. We *cannot transform life* without an anchorage in the forces that go through whole lives (including the continuum of the *vida mula*), as Juguetes Perdidos shows:

> A powerful militancy is that which, beyond the "programmatic", seeks unexpected alliances that transcend the external and internal frontiers of the new [impoverished] neighborhoods; a militancy that can think about the burning problems and the dirty workarounds, about the ways in which a life and a death can be politicized (increased in value) . . . A militancy that can withstand what is ambiguous and unmoral in the forces that go through concrete lives . . . (which are neither heroic nor saintly). [21] (p. 59)

In order to *change life*, a way to break free from the chains of habit must be found, without appealing to a sacrificial moralism such as Selci's. What it takes is a *sensible offensive*.

## 3. From the Pathogenic Situation to the Transforming Action

While it is true that every political action, as we have seen above, must necessarily have an impact on the *extensive field* (by taking care of the material needs of life in society and proving *useful* for popular struggles), this field is not enough, as the common sense would have us think. It is not enough because we are trapped in the automatisms that form a pathogenic milieu and reproduce precarious ways of life. Lives without food *or* sense. A political construction cannot aim to improve the lives of the majorities without casting off the pathogenic milieu that perpetuates the present situation, and this cannot be done without resorting, in the first instance, to the *intensive field* and the *individuations* that populate it. In our exposition, then, we should move towards the second of the three fields that compose Deleuze's immanence. There we can find the germs of an action that can enable the form SAS′, that is, that produces a *modified* situation (S′). Briefly, even if the action-image places itself in the extensive field (given that the situation is "a set of power-qualities as actualized in a milieu, in a state of things or a determinate space-time" [2] (p. 142)), that milieu can only be transformed—and therefore be a full part of the action-image—insofar as it makes room for the affections and the potencies that are embodied in it [2] (p. 141).

The milieu is only one of the two poles of the action-image [2] (p. 142) and the situation is only one element in the SAS′ form. We have not yet considered the second pole: the duel. At first glance, the duel does not seem to let us go any further into our problem, considering that it also appears to be placed in the extensive field. It "involves in its very exercise an effort to foresee the exercise of the other force", since "the agent acts as a function of what he thinks the other is going to do" [2] (p. 142). Therefore, this "act" remains prey to habits and regularities that make possible the "feints", "parries" and "traps" that are part of any duel [2] (p. 142). Even if Deleuze applies them "*par excellence*" to the final duel in Westerns, with the cowboys in the dusty streets and the guns at the very edge of being drawn, the terms "feints", "parries", and "traps" ("*faintes*", "*parades*", "*pièges*") belong to the vocabulary of fencing, a sport with a long tradition in France. Let us think about a simple duel between two fencers. The fencer trains in order to foresee the attack of the opponent, to act as a function of what he thinks the other is going to do. Nothing is

spontaneous; the feints, the parries, and the traps are part of the training, which is nothing else than the acquisition of the accurate habits for a better performance.

However, most athletes would say that at the moment of the bout there is something that goes beyond extensive time-space. Something prior to the embodiment of the movements that will define the duel. Imperceptible, unceasing. An *intensive* space that pervades the seemingly close milieu where the action must take place, and makes it possible for the blade to pass, and for the point to smash against the adversary's breast. Later, all seems to have happened in the extensive space. Victory is celebrated, the points are counted. Nonetheless, that fencing hit, in the very same way as a soccer pass that slips several opponent defenders to meet the foot of a partner just in front of the net, can only have taken place in another space-time: intensity. "The specious present feels this intensity of thinking pass into action. Normally the intensity itself is overshadowed by the effectiveness of the action it passes into" [23] (p. 67).

This mandatory passage from extension to intensity can be observed in the *élan* that pervades the chapter on the action-image in *The Movement-Image*. Indeed, even though the first impression when we read this chapter is that we are always in the extensive field, the structure of the text only makes sense if we take into consideration the play between both fields: the extensive and the intensive. The first lines refer to the *affects* that are actualized or embodied in determinate space-times. Later, the argumentative arc leads to the very same issue that we are trying to resolve here: Deleuze asks himself how "the passage from S to S'" is organized [2] (p. 151), and then he answers: through the situation *as well as* the action (they have this common fate: there is no transformative action without a transformation of the milieu, which at the same time can only be transformed by the action). In the first case, it happens by "the division of the principal situation into secondary situations which are like so many little local missions within the global mission" [2] (p. 151). In the second case, "through the intermediary of A, the decisive action", the milieu "must contract into a binomial or duel in order for the powers which it actualizes to be redistributed in a new way" [2] (p. 152). The duel, which had already been mentioned in the first pages of this chapter, reappears as the protagonist in the passage from S to S', from the situation to the transformed situation. The action disengages from the situation of origin as the duel takes on another dimension, first as the contraction of the situation, later as the duel in itself, then as a "dovertailing of duels" which gradually fills the "big gap" between the situation and the action, allowing the "hero" to be finally capable of an action that was "too great for him" before [2] (pp. 153–154). The duel is also mourning (*deuil* in French): a process that allows us to break free from the attachment to the pathogenic situation, an attachment that is often melancholic, and sometimes even self-destructing. There is attachment to the *vida mula*, there is libidinal investment of servitude, as a response to the key question of political philosophy: "Why do men fight for their servitude as stubbornly as though it were their salvation?" [13] (p. 29). One fights for servitude but undertakes mourning in order to transform the situation.

Then, once again, how is transformative action possible? "On the one hand the situation must permeate the character deeply and continuously, and on the other hand the character who is thus permeated must burst into action, at discontinuous intervals" [2] (p. 155). Deleuze offers more clarifications, but the doubt persists: how is it possible for a potentially pathogenic situation to permeate the character? How is it possible for the permeated character to burst into action and not relapse into the *vida mula*? A fleeting hint shows us the way: the structure formed by the couple permeation-bursting is "that of an egg" [2] (p. 155). The egg leads us again into *Difference and Repetition* and the *intensive field* that makes individuation possible: "Individuating difference must be understood first within its field of individuation —not as belated, but as in some sense in the egg" [17] (p. 250). The egg is a field of intensities, "haecceities" that have "the individuality of a day, a season, a year, a life (regardless of its duration)—a climate, a wind, a fog, a swarm, a pack" [1] (p. 262). If the egg is the structure that permeates the situation and forces it to burst into a transformed situation, in these winds lays then the possibility of emancipation.

They are the transcendental element (condition of possibility) of the action structure: SAS′. The possibility of a transformative action is born.

## 4. Intensive Action: Sensible Offensive and Wild Life

"An intensity is not something philosophical or abstract, an intensity is something that fleets normality, something that pierces Reality (*vida mula*) or better, something that fleets that continuum" [21] (p. 82). Extension does not exhaust existence. There is another field, another truth, the "glaring, somber truth that resides in delirium" [8] (p. 4), without which there would be no life: intensity. In the words of Brian Massumi: "Intensity does not 'have' value. Intensity is a value, in itself. In fact, it is a surplus-value: a surplus-value of life. It is a more to life, in life" [23] (p. 99). The *continuum* is full of holes, and through those holes the intensities that every extension envelops flow endlessly. Steam in the pores. A heat that no increase in entropy can dissipate. Many actions respond to that stimulus and awaken such a sensibility, but, due to their very nature, they are evanescent. Fragile and precarious as the ground on which the *continuum* of the *vida mula* is based. Suddenly, they awaken a smile or insufflate a breath of fresh air, even produce a decrease in entropy (and thus they momentarily counter degradation). Suddenly, they rise in intensity and have a bursting effect. Sometimes it is individual: outburst of anger. Sometimes it is in groups: street violence. Sometimes it is collective: the streets filled with colors and popular rebellion.

It is no wonder then that intensity (and the constellation of terms connected with it, such as affect, sensibility, event, becoming, micro-politics, etc.) is the field *par excellence* of Deleuzian political studies. This can be seen all over the globe, as a recent book from Cambridge Scholars Publishing exemplifies: "Reyes narrates to us the possibility of understanding revolution from the point of view of small politics, that is, micropolitics. . . The possibility of a revolution, normatively based on a vague notion of freedom, is brought about by temporal, albeit non-sequential, moments. This means that the 'micro' moments of micropolitics come from different directions and in various degrees of intensity" [24] (pp. ix–x). Massumi shares this take on Deleuzian political philosophy: "This way of thinking about politics in terms of contrasts and lived intensities of feeling" [18] (p. 100). In the French-speaking world, Sibertin-Blanc underlines minor struggles as the place of political subjectivation [25]. In these pages, we will focus on the South American scholars who share this perspective. The Hungarian-born Brazilian philosopher Peter Pál Pelbart stresses *the common* as a "reservoir of singularities in continuous variation", capable of making individuations possible, and which shapes a realm of resistance through "redistributions of affect" that lead to "the new possible" [26] (p. 24). The kind of action that Pelbart seems to have in mind is thus an opening to a virtual chaos that may pierce ("in each pore") the capitalist captures. To act is to get rid of those particularities that "oppose men against men", but this does not equal fusing them in a single whole: the "originality" of each individuation remains, in the sense of "a sound that each of us *utters* when he sets a foot on the road, when he leads his life without seeking salvation, when he sets out on his embodied journey with no particular goal" [26] (p. 39). There is a clear affinity between Pelbart's theory and the intensive perspective: the *continuum* must be pierced, and the chain of current and actual particularities broken, in order to create new possibilities. Only then may appear what the *vida mula* had sealed: the life of singularities and *haecceities* (the individuality of *that* utterance). It is a matter of *fleeing*: "in each interruption of the flows there is a point of possible detour (flight ["*raje*" in Spanish slang]); an interruption that may either be immediately axiomatized, coded and inserted again by morals, or imply a powerful leap of role, a liberating leap of scene, an experimentation" [21] (p. 58). In spite of capitalist realism, new individuations open up.

We can still ask: which is the "transformed situation" that these individuations open up? Fleeing and riddance are not enough, because they expose us to the risks of falling into the abyss, of destruction or, in "the best scenario" of rediscovering the forces that compel our body, in this very present, in this very instant in which we live, to go back to the *continuum* in a universe of precariousness. It takes a *political action*, one that is no

longer individual, not only a flight, not only a riddance, but a *construction*, a *creation*. This is the core of Brazilian theorist Suely Rolnik's claim, when she focuses upon the social movements that were born in the 90s: "The *intensity* with which those movements emerged . . . tends to bring about a temporary destabilization of its [the colonial-capitalist regime's] tyrannical omnipotence" [27] (p. 26, my emphasis). The significance that this Brazilian philosopher gives to the intensive field is so large, that she even coins the term "intensive resonance" to characterize "the micro-political sphere of human existence; inhabiting it is essential to situate ourselves in relation to life, and to make the choices that can protect and boost it" [27] (p. 101). This stress on *intensity* can also be seen in the precarious and evanescent nature of the social movements: "as fast as they appear, they vanish, only to reemerge again immediately" [27] (p. 101).

In a footnote, Rolnik admits that *intensity* has not only micro-political dimensions but also *macro*-political ones (some social movements work exclusively in one of those dimensions, others in both), and she even protests against the "pernicious dichotomy between micro- and macro-politics" [27] (p. 131). Nonetheless, the relation with macro-politics is always a *demand* directed at an instance that remains exterior to the people. Rolnik allows that the law and the institutions are part of the improvement of the quality of life, and even that this improvement requires the protection of the Apparatus of Capture (which are mostly related to the State form). However, she never posits an organic relation between the State and the micro-political movements (we will return, in our conclusions, to the notion of an Organic State that does not contradict the Deleuzian spirit). The macro-political aspect remains secondary and inadequate: "It is not enough to resist to the current regime macro-politically, it is also pressing to work in order to reappropriate the force of creation and cooperation —that is, to act micro-politically" [27] (p. 30). In Rolnik, the question about how to *transform life* returns in terms of "pimping" (a term with its own conceptual specificity, that we will not study here): "How do we liberate life from its pimping" [27] (p. 34). The answer (the *action*) will be "the production of subjectivity, desire, thought and a relation with the other that leads us to surrendering blindly to the appropriation of the force of creation" [27] (p. 34). It is a question of the position of *desire*, of complying with or turning away from the *discouraging* present situation (S): a colonial and capitalist unconscious [27] (p. 52). Between liberation and servitude there is a micro-political, that is, intensive conflict (taking into account that the weapons of contemporary capitalism are also, and mainly, micro-political).

All these perspectives find their most neat expression in the title of one of the books from another Argentinean Deleuzian philosopher, Diego Sztulwark: *The Sensible Offensive*. "If thinking otherwise requires feeling otherwise, the battle of ideas should be preceded, or at least joined by, a *sensible offensive*" [28] (p. 26). This offensive considers that the task of philosophy is to diagnose becomings and to acquire knowledge of the affects [28] (pp. 177, 180). The place of political creation is, according to Sztulwark, the intensive and micro-political field, while macro-politics is restricted to the preservation of what is given [28] (p. 21).

In the sensible field, a new way of life can be found, which can empower our conditions of existence as human beings. It is what Juguetes Perdidos calls "wild life" (*vida silvestre*):

> The wild as intensity makes another calculation of precariousness-consumption-waste-work, it puts together another series (or tries to put together another series) with those elements, disrupting certain molds and moving in a new fashion before the cliff. [21] (p. 35)

Briefly, the intensive field is effective at bringing affects to their boiling point, spreading the cracks, revealing the miseries of the capitalist life and its levels of suffering that sometimes remain imperceptible, hidden beneath the common sense. However, in order to *transform the situation*, a dimension that is determinant and constitutive of new *means* is required. It takes a creative act, and this demands *having an idea*.

### 5. Ideal Action and Creation

Political creation, in the very same way as artistic and philosophical creation, indeed demands that one "has an idea". To reach that conclusion, it suffices to reach the full implications of Deleuze's conference "What is the Creative Act?":

> What does it mean to have an idea in cinema? If someone does or wants to do cinema, what does it mean to have an idea? What happens when you say: "Hey, I have an idea?" Because, on the one hand, everyone knows that having an idea is a rare event, it is a kind of celebration, not very common. And then, on the other hand, having an idea is not something general. No one has an idea in general. An idea —like the one who has the idea— is already dedicated to a particular field. Sometimes it is an idea in painting, or an idea in a novel, or an idea in philosophy or an idea in science. And obviously the same person won't have all of those ideas. Ideas have to be treated like potentials already *engaged* in one mode of expression or another and inseparable from the mode of expression, such that I cannot say that I have an idea in general. Depending on the techniques I am familiar with, I can have an idea in a certain domain, an idea in cinema or an idea in philosophy. [16] (p. 312)

In this lecture, given in 1987, Deleuze recovers the main concept of *Difference and Repetition*, published almost 20 years before: the Idea (even if the capital "I" is left behind in the transcription). The consequences of this gesture for political philosophy are vast. Deleuze says: *Sometimes it is an idea in painting, or an idea in a novel, or an idea in philosophy or an idea in science*. I add: sometimes it is an idea in *politics*.

Extending the series of acts of creation to politics implies a new take on the *social Idea* as thought by Deleuze in *Difference and Repetition*. Of right, Ideas are multiple, and should apply to any political action; in fact, however, Deleuze just offers a restricted image, where social Ideas are not only macro-political, but specifically great structures whose weight crushes any human gesture or action, be it individual or collective (social Ideas are the germ of the *forms of socius* and the Apparatus of Capture that he will develop in his works with Guattari). Selci's objection, which we quoted in the introduction, repeats itself: "The enemy was so big, and the character of the emancipatory struggle so 'local' and small, that philosophy seemed to be content with supplying a sophisticated theory for lucid resignation" [14] (p. 11). In *Difference and Repetition*, the social Idea is certainly typified in terms of capitalism: "[The social Idea] expresses a system of multiple ideal connections, or differential relations between differential elements: these include relations of production and property relations which are established not between concrete individuals but between atomic bearers of labour-power or representatives of property" [17] (p. 186). Later in *Anti-Oedipus*, this germ is extended to the *forms of socius* or social machines: the primitive territorial, the barbarian despotic and the civilized capitalist [13] (pp. 139–262). Finally, it acquires its most sophisticated exposition in *A Thousand Plateaus*, where cities and worldwide organizations are also studied [1] (pp. 375–403). They are all Apparatus of Capture, which can only be resisted by War Machines (a more developed version of this argument can be found in [6,29]). However, in the present situation, these machines fall under the power of capitalism, and hence the situation becomes highly discouraging [13] (p. 422). Nowadays, only one social Idea is embodied, that of Capitalism, which determines the deepest logic of our common being. Thus, the web from which we cannot free ourselves is spun. *It is easier to imagine the end of the world than it is to imagine the end of capitalism.*

However, the Capitalist global War Machine in a *macro*-political level does not cancel the working of the nomadic war machines in a *micro*-political level, that is, in the intensive field. It is no wonder that, as we reckoned above, the most eager political Deleuzians try to flee through that gap. We are suffocating under the folds of the Capital, *we need to undo them*. We must flee, deterritorialize, smooth the space whose striation is asphyxiating us (just to mention some of the concepts that Deleuze has created in order to think this peculiar process). We should "undo the doubling and pull away the folds", as Raymond Roussel

did in literature [30] (p. 99). However, it is precisely there that a problem emerges that Deleuze himself noticed: "at that point you go back to the unbreathable vacuum . . . You undo the folds and you spread them apart, like a swimmer, you return the sea to itself and you die" [31]. How do we explain this phenomenon of asphyxiation in the vacuum? Because they are affects, becomings and intensities *without Idea*. For this very reason, the nomadic working of the war machines only offers the possibility of flight, resistance or—at the most—brief and ephemeral constructions. By restricting the social Idea to the great structures and Apparatus of Capture, Deleuze rendered all constructions that men try to build on the ocean of precariousness "Idea-less". He left minorities at the mercy of the social elements, exposed to the violence of the strongest.

Hence the great impact that the introduction of the Idea in the lecture on the creative act had on political philosophy, and this is because, when Deleuze thinks about painting, about literature, about philosophy or about science, he poses a multiplicity of Ideas. A large variety that is not reserved only to a fistful of "great men", but is available also to those that remained in the shadows of history, those who "lost" the disputes in which they were involved, those who are not included in any canon. It is not only a matter of revolutionary Ideas, it is not only a question of inflection points. There is a multiplicity of creators for a multiplicity of Ideas. Some of them had only one Idea. Some of them had so many Ideas that we never get tired of exploring them. So many philosophers, so many film makers, so many artists. Not only the great movements, not only geniuses. Nonetheless, when it comes to politics, Deleuzian Ideas seem to be reserved exclusively for the great civilizations, as we showed above.

Thinking about the social Idea as an act of creation at the very same level as cinema, literature, or philosophy means opening that field to *minor* creators who nevertheless create Ideas and, therefore, do not belong to the *intensive* field, but to the virtual. From the ontological point of view, the Idea is the concept that determines the virtual field, through differential relations and the repartition of singularities: "Ideas thus defined possess no actuality. They are pure virtuality. All the differential relations brought about by reciprocal determination, and all the repartitions of singularities brought about by complete determination, coexist according to their own particular order in the virtual multiplicities which form Ideas" [17] (p. 279; the characterization of Ideas as virtual can also be found in pp. 191, 207–209 and 284). The Ideas are virtual machines that produce the determinations that make something what it is. They are not—as in other philosophical frameworks—fixed determinations, but a determination that endlessly changes and becomes. This variation does not contradict determination, but constitutes it: we would not be what we are if we were fixed, stopped, because what we are is, properly, a variability. This machine works through three gears that Deleuze develops in the chapter of *Difference and Repetition*: the indeterminate ($dy$, $dx$), the determinable ($dy/dx$), and the determination (the values of $dy/dx$). These formulas show that Deleuze uses mathematical concepts in order to construct this concept (a comprehensive reconstruction of Deleuzian Ideas and their mathematical background can be found in what should be part of the mandatory secondary literature on Deleuze, Gonzalo Santaya's *El cálculo transcendental* [32], which was partially translated to English [33]; other substantial secondary literature that covers the notion of virtual Ideas are Williams ("Virtual ideas are relations of all pure becomings" [9] (p. 8) and Smith, who recovers the crucial relation between Deleuze's Ideas and those of German Idealism [8]).

If we keep in mind this ontological aspect of Ideas while considering the "ideas" that Deleuze mentions in his lecture on the creative act, and we go beyond Deleuze's argument to extend the realm of creation to the political field, then we can develop the full impact of Social Ideas. They are capable of producing *new determinations* in actual existence. They are sometimes thunderous and *major* (the big revolutions, the civilizational ruptures, the emergence of movements that change the history of a country or a region, such as Peronism in Argentina in 1945, or Zapatism in Mexico in 1994), but usually *minor*, more humane, within reach of fragile actors, leaders, and political militants who populate the actual

institutions whose actions are decisive for daily affairs. The figure of the Politician, so degraded by the social agenda, acquires a new sense, a new dignity.

Through Ideas, determinations and consistencies are weaved. They are not ready-made, they do not lie prepared in the hand of a god, nor are predestined by a logic that leads the way in the process of the Real. They are not transcendent: they do not act outside or above the intensities and the extensions that we have outlined before. Ideas contain in themselves the indeterminate, the ontologically most precarious, what is in itself nothing else but a shadow behind our backs, but it is also in Ideas that the indeterminate determines itself and becomes a machine of production of singularities. These determinations give consistency to the intensive processes that billow through the globe. They make it possible for the wild sprouts that emerge in the slums to acquire a voice, a space, duration, and thus build a precarious territory in the midst of the *vida mula*. There are "other liturgies to deal with the intense and also to deal with totalitarian precariousness; in order to manage the balance of the moods, the joys and also the nausea, the collapses. Other images of what it means to live and die" [21] (p. 86). Ideas are necessary if these other images are to last, to have determination, and the strengths to open a new world in the midst of the totalitarian drive of capitalism.

To be the only Idea, to reach the One in the world: that is the drive of the capitalist logic of the social. To multiply the Ideas is the way to transform what is pathogenic in the milieu, and to be able again to have illusions about our society and human nature.

## 6. Conclusions: The Macro and the Micro

To have an Idea is something rare, but at the same time within reach of everyone, in different measures and magnitudes. The Idea *determines* an action that can change the world, precisely because it does not take its determination from the present situation or the milieu as ready-made. It does not respond to the *continuum* of the *vida mula*. A film maker takes the camera or goes into the cutting room; a painter faces a canvas that is full of *clichés*; a philosopher searches for a concept in response to a problem that does not let him or her sleep by night. The politician (from the president of a nation to a militant of the smallest organization, from the one that signs the decrees to the one that hands out fliers and pins) is not very different. Judging the figure of the politician by the empirical examples, the failures, the betrayals and the impotence, is the same as judging movies, literature, or philosophy by the failed sequels of the worst pop-corn films, the novels that are forgotten as soon as they are read, the concepts of no interest. There is no impotence *by itself*. The politician *can* have an Idea and break the *continuum*, the sentence to live as we do, the subjugation to these gray rules, the complying with the dominant powers of the world and with those who wield them as if they owned them. It is not a question of dreams or convictions. It is about Ideas. About the just Idea, which will never come out of inspiration or rapture, but of listening to the voices of all those intensities that the uproar of the *vida mula* will not let us hear. "We have to think, make a cartography, research, be militant, stir up and withstand those silences that prevail until the sounds, the screams and the whispers, the monologues and the murmuring begin to be heard" [34] (p. 64).

To move forward in this path, it is necessary to dismantle the double bind that menaces the political thought of Deleuze and Guattari: the one between the micro-political and the macro-political. It would seem that the macro-political is the field of the social Idea and its embodiments, where the Apparatus of Capture, the disciplinary societies and the societies of control reign. According to this interpretation, on the one hand, Ideas seem to be the origin of rigid segmentarities and molar organizations; on the other hand, the extensive *is* macro, and therefore condemned as such in the name of molecular revolution. There are many passages where Deleuze himself holds this position:

> Between macro- and micromultiplicities. On the one hand, multiplicities that are extensive, divisible, and molar; unifiable, totalizable, organizable; conscious or preconscious —and on the other hand, libidinal, unconscious, molecular, intensive multiplicities composed of particles that do not divide without changing

in nature, and distances that do not vary without entering another multiplicity. [1] (p. 33)

The problem with this position is that it is axiological: it poses a dualism where good and evil are already distributed. An unbreakable evil alliance is forged between Ideas and the extensive, which dooms all our actions in this field to reproduce unconsciously the laws and modes of life of the forms of socius: "existences that are deduced in a direct fashion from the bonds that the rule of the capital proposes" [28] (p. 38). In the best scenario, from the extensive can emerge institutional claims that *do not really change anything*; it is what Deleuze and Guattari called the axiomatic struggle, where basic demands such as land, roof, and job, or rights as equal marriage, laws of gender equality, and legal and safe abortion are played down; as if improving extensive life was always the same as playing for the Capital [1] (pp. 470–471). As a consequence, micro-politics and macro-politics oppose each other, one is the *contrary* of the other: "Becoming-minoritarian is a political affair and necessitates a labor of power [*puissance*], an active micro-politics. This is the opposite of macro-politics" [1] (p. 192). The micro is the "good" option and, as Patton has pointed out in this classic *Deleuze and the Political*, has *priority* in relation to the macro: "Deleuze and Guattari treat rhizomatic, molecular and micropolitical assemblages as prior to arborescent, molar and macropolitical assemblages, and the abstract machine of mutation as prior to the abstract machine of overcoding" [35] (p. 45). This priority is axiological, as well as ontological. This way, Ideas appear as the "evil" option, while the "good" one is the "forms of life" that are able to short-circuit the automatisms [28] (p. 38). Micro-politics means resistance, tracing intensities, and supple segmentarities in "multiple molecular combinations" [28] (p. 260). As we have already seen above: micro-politics is that which flees and breaks, but never gives way to stable constructions, because these obey a logic that contests and betrays them.

This axiological interpretation of the letter of Deleuze can and should be put into question. I believe that, even if it has less textual support in his work, giving full importance to *both* the macro *and* the micro dimensions is more productive in order to think what is possible in the difficult times we live in. I believe that macro-politics without micro-politics is blind, and micro-politics without macro-politics is empty. Intensities without Ideas are lost in their own evanescence and fall into a *vida mula* that gets rougher and rougher; while giving up the extensive leads us to a politics that neglects the urgencies of the dispossessed and the protection of the minorities that are still hurt after a long history of injustice (intensities are "out of this world", Peter Hallward would rebuke [36]). To get out of this dead end, all the conclusions that derive from this assertion: "every politics is simultaneously a *macropolitics* and a *micropolitics*" [1] (p. 213) must be fully deduced. This does not mean that "they are the same", because they "do not envision classes, sexes, people, or feelings in at all the same way" [1] (p. 196). However, the question should not be restricted, as Deleuze and Guattari believe, to mutual interferences and reactions between the two lines [1] (p. 196). In order to rise to the challenge of Deleuzian ontology, we must be much more ambitious. It is a matter of *immanence*, of considering in different ways what is univocal, of taking different paths for the one voice of being. *There is no micro without its unity with the macro.* There is no such thing as*: first the macro, then the micro, finally their unity.*

It is easier for Deleuze and Guattari to admit the univocity of the political when they consider the micro-political aspect of molar organizations: "The administration of a great organized molar security has as its correlate a whole micro-management of petty fears, a permanent molecular insecurity…: a macropolitics of society by and for a micropolitics of insecurity" [13] (pp. 215–216). However, they are incapable of the reciprocal reasoning; they cannot envision a macro-political aspect of the "molecular movements", and insist only on their capacity for resistance and for "thwart[ing] and break[ing] through the great worldwide organization" [13] (p. 216).

> Politics operates by macrodecisions and binary choices, binarized interests; but the realm of the decidable remains very slim. Political decision making necessarily descends into a world of microdeterminations, attractions, and desires, which

it must sound out or evaluate in a different fashion. Beneath linear conceptions and segmentary decisions, an evaluation of flows and their quanta. . . . Good or bad, politics and its judgments are always molar, but it is the molecular and its assessment that makes it or breaks it. [13] (p. 222)

In order for resistance to become political construction and for flight to take us to habitable spaces, it is necessary for the micro to find its macro dimension. Only thus will our creations persevere and last, which is not at all absurd if, besides the way in which the macro becomes micro to dominate and subjugate, we consider how the macro *is* also micro in order to establish alliances with the sensible offensive. As a matter of fact, the politician is a statesman, but also (and as such) must dive into the molecular that makes the political. In this point, Deleuze takes us beyond the classical representative politics. Without these micro-determinations, the action will fail (in no definite axiological sense: it can either fail in dominating and subjugating, or in empowering and nourishing). The State is a rheumatic elephant, and is used to one Idea, that of Capitalism. However, that is the macro perception in a pathogenic milieu, where the decision is restricted to conforming to the requirements of the global market or resisting through controls, taxes, barriers, and restrictions. As if there were only two Ideas in struggle: on the one hand, a fake State, without sovereignty, subservient to the capitalist Idea; on the other hand, an authentic State, sovereign, self-regulated and transcendent (divided from the people, their extensions and intensities). Underneath that polarity, there is a world of flows and quanta, of affects, intensities, and *haecceities*. Through a series of duels, the politician is able to mourn both the dream of a transcendent and perfect State and the resignation to a State that only serves capitalism, and dive into the intensities that point toward a new social realm: that which the editors of the Argentinean journal of philosophy *Ideas, revista de filosofía moderna y contemporánea*, with a Deleuzian spirit mixed with some Idealism, have called an "organic State":

What do we understand by the term "organic State"? What does "organic" mean in this expression? How should we understand this dangerous metaphor, potentially laden, for example, with closed functionalism, natural hierarchies, submission and subsumption of a part under the whole, or with fascism? In the first place, and leaving aside those connotations, organicism means a vital intertwining of the parts so that what affects one of them, affects also the others and the whole. [. . . ] Not all parts of a living body must be thought of as if they were organs. Indeed, biological organisms have components that neither are nor belong to organs, as for example the bacteria that can be found in the digestive system and form a paradoxical inner exteriority. Every organism is inhabited and traversed by a multiplicity of inorganic elements, without which it could not live [. . . ] In the same way, the logic of the State is not the only thing that moves social processes as if it were the first mover. There is a whole dynamic between the organic and the non-organic components, molecular exteriorities and topologies that are not necessarily synthetized in a unity. [37]

In the same line of thought, I have been working on the concept of a Deleuzian State, which, in contrast with the classical conception of the State, does not have a transcendent ground as an eminent Unity (embodied in a Sovereign or man of State), but emerges from a logic a multiplicity and variation, in the very same way that the Deluzian Idea rewrites the history of the Idea, from Plato to German Idealism [38]; in these pages, I follow that research by thinking the kind of politician that can replace the Sovereign. The aim of this line of work, in these difficult times, is not only to break through the dominant Idea in the actual (capitalism), but also to produce Ideas in the frame of a multiplicity, solving local problems, creating precarious or stable spaces, sheltering the minorities, and giving voice to that which was left invisible and suffering. All these micro-political flows can find in the macro-political their empowerment and harbor. Once again, macro-politics is not external and does not transcend micro-politics; radical immanence means that macro-political Ideas do not exist without the intensive and extensive fields that are part of their

genesis, determination, and existence. In other words: minorities are an organic, active component of the State, of its intensity, sense, and value. The politician is the one who can have that Idea (again, sometimes monumental, sometimes small) and move from the present situation (S) to the transformed situation (S′), because he is able to connect with an intensity that, until then, had been Idea-less. The politician is the one who can hear the clamor of the people, for better or for worse, because he is in himself a minority and part of the sensible offensive (once again, there is no axiology here, there are political Ideas that bring new forms of suffering, domination, and exploitation, as well as political Ideas that heal, feed, shelter, empower, and free). Through Ideas that only he can create, the politician has therefore the key role of opening new lines of creation, even within a highly discouraging situation, in order to transform, at one and the same time, life and the world.

**Funding:** This research was funded by National Scientific and Technical Research Council (Argentina): 140780.

**Institutional Review Board Statement:** Not applicable.

**Informed Consent Statement:** Not applicable.

**Data Availability Statement:** Not applicable.

**Acknowledgments:** This article was reviewed by Matías Soich, who contributed both to the technical aspects of the English language and to the precision of the philosophical arguments and concepts. All the translations from the Spanish have been made in collaboration with him. Rafael Mc Namara has provided invaluable advice regarding the use of the term "entropy" by Gilles Deleuze, which he had thoroughly worked in his book *La Ontología del Espacio de Gilles Deleuze* [39].

**Conflicts of Interest:** The author declare no conflict of interest.

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
