# Peer review of "The Politician: Action and Creation in the Practical Ontology of Gilles Deleuze"

_philosophies, doi:10.3390/philosophies7030050_

Round 1

Reviewer 1 Report

This is an interesting and important article in at least two dimensions. First of all, it aims to show how Deleuze's political philosophy should be understood as practical philosophy and as something which also helps in political action and activity, and not only in explaining the phenomena occurring in political sphere. Thus, it takes seriously old Marxist agenda given for philosophy - the aim is to change to world, not only to explain it, which I find important. 

Secondly and more particularly this article aims to show the interconnected "nature" of macro- and micropolitics (or major and minor etc.). This also is important, since Deleuze's political philosophy is sometimes interpreted, perhaps sometimes even against Deleuze's own understanding, building on ontological dichotomies, which sometimes are even seen as excluding one another. In this article, however, the authors offer an understanding of micro-macro as complementary, not excluding. 

As I see it, the author(s) succeed rather well in the first task, but the second is left more as a claim, which is not elaborated succinctly enough. The idea and argument comes through, though, so there are no absolute need for serious rewriting / elaboration of the text. Still I feel, that much more could have been said about micro-macro relation and the way both of these aspects are not only found in the activity of political sphere, but also needed. However, I realize the difficulty of going deeper to the interconnections of micro-macro in the limits of one article. 

In the current form the basic idea and argument of the article is sound and clear. Or perhaps: it is sound and clear for the readers of Deleuze and Guattari, but is it for the others? This is my main critique for the article as it stands: it contains a good and coherent work on some of the Deleuze's main concepts in relation to perhaps less analyzed concepts such as "activity" etc. and manages to combine and produce a text that really offers a good and sound aspect to Deleuze's political philosophy. However, every here and there the authors use Deleuze's concept's assuming that the reader knows them rather well. Concepts are not really explained or defined - which sometimes is indeed a difficult to do shortly in Deleuze's case - and thus the reader _must_ know what is rhizome, what is the content of the difference, or  to be interested to check Mille Plateaux to find out what ritornello might mean.

In other words, for the readers or Deleuze, this article appears as rather sound and down-to-earth elaboration of Deleuze's political philosophy (which I like). However, the authors might also pose a question of the other audiences than only the (likeminded) readers of Deleuze: how does the content and basic idea of the article translate to someone who does not really know her Deleuze, or is only a beginner with Deleuze and political political philosophy in general?

Of course, the aim of the scientific journal article is not to produce textbook level introductions or "Deleuze pour les nuls", but on the other hand, this article has, in my opinion, also a message that would be important for the wider audience than just Deleuzians. And, this is even more important because the article is connected to South American political thought (which acts really well in dialogue with D&G) and even the South American cases and political field / milieu. For this reason I advice - but not demand - to clarify the article a bit more: to open up the central concepts, to offer references to secondary literature on Deleuze and Guattari, to tell how you understand these concepts and what other possibilities there would be. Perhaps it could be also wise to cut out a little bit of long citations (and even references) to many different authors and concentrate more on the main characters (Deleuze, Guattari and Juguetes Perdidos). Perhaps, add more of your own explanations, and not use so many citations. On the other hand, as I stated before, a little more references to the secondary litterature on D&G might be useful (there already is Massumi, for example, but there could be more). 

In general, I find this article interesting, good and refreshing interpretation of Deleuze's political philosophy. English is rather good, but of course it is always good to check the language at the end. 

Author Response

First of all, I would like to thank you for the attentive reading of my text; I’m glad to know that you consider my article important and interesting. I understand that you find two main points that may be improved: in the one hand, I could go deeper in the relation between macro and micro politics; on the other hand, there is obscurity in the introduction of some of Deleuze’s concepts. I agree with you in both objections and, even if you feel that there is not absolute need for serious rewriting in any of the cases, I will work in the final manuscript in order to improve both aspects, that may prove to be interconnected. By clarifying some concepts, I may show how deep the relation between macro and micro-politics in my argument – and by developing a littler more the question of macro and micro-politics some concepts will gain in clarity. I will spare some concepts that bring more obscurity than sense (one of the cases that you point out, the ritornello, demands developments that I have made elsewhere, and that are not of the essence here), and will clarify briefly some of the key concepts, in order to reach, as you say, a wider audience (taking care not to over-simplify). I believe that this will allow the final manuscript to reach its objectives with more strength. I thank you again for your insights, and I hope you will be satisfied by the final version.

Reviewer 2 Report

Referee report for "The Politician: Action and Creation in the Practical Philosophy of Deleuze

This paper uses the philosophy of Giles Deleuze (particularly his reflections in Cinema 1 and What is the Creative Act?) to argue for the creative role of the figure of the politician. It draws on a range of Latin American (mainly Argentine) Deleuzian philosophers, and it is also reasonably well written (especially given the remarkable obscurity of Deleuze's terminology).

It is not clear to me what the contribution of the paper is. On one reading, the paper just presents the politician as a figure that can have a transformative "Idea" that bridges the micropolitical and the macropolical, and is thus able to transform a situation. But this appears to me to be a truism, even if it is not widely accepted by the community of Deleuze scholars. Little is said about the nature of these ideas, except that they resonate with the intensive field, and need not be the "grand gestures"(Capitalism, the State, etc.). I confess I have little sense of what these ideas may be; the paper's highly abstract register does little to help.

The concern with the creativity of the politician - as someone who can act in ways that transform a situation - seems to me to be misplaced. We know from Arendt that all action brings something new into the world; we do not need to draw on Deleuze's reflections on cinema to know that political action can transform a situation. And the general framework - of the politician who is able to break with the structures of capitalism - seems derivative of Weber's reflections on charismatic authority in the "iron cage" of modern rationalization, though using a very different terminology. 

In any case, the way the paper is framed, the politician does not merely act, but is able to have "ideas" that go beyond the dominant "idea" (capitalism). The paper says little about the sources of these ideas - the everyday life of individuals perhaps? - and very little about their value. How do we distinguish transformative ideas from ideas that just reinforce dominant narratives? Are some of these not really "ideas"? Worse, how do we distinguish between ideas that are creative in a positive way, and ideas that are creative in a very different way? The bracketing of the question of value (only briefly raised at the end, where the author notes that "there are political Ideas that bring new forms of suffering, domination and exploitation, as well as political Ideas that heal, feed, shelter, empower and free") does not help here; political action is inescapably about changing situations, and so the key question is more about responsibility (as Weber saw) than about creativity.

There is a long tradition of seeing the politician in the model of the artist, and that tradition leads to no good; the author should be clearer in any case about the difference between the politician and the artist. There is perhaps an interesting germ of an argument here about how the politician articulates something of the lived intensities of a situation through their representative work, but nothing is developed in detail; at the end of the day we are left only with a number of truisms about the difficulty in bridging micropolitical resistance and macropolitical action, which this paper does little to resolve. 

Author Response

First of all, I would like to thank you for the attentive reading of my text. It is always enlightening to have a new perspective in some ideas I have been developing during a lengthily period of time. I will work hard on the final version of the manuscript in order to try to make it clear that the contribution of this paper consists precisely in the tension that you are pointing out: I present a “truism” that is not widely accepted in the community of Deleuzian scholars. What I believe you are wondering is why should I use the concepts of Deleuze in order to argue a truism that can be easily supported by Arendt, Weber and so many others (to give another example, that I have worked somewhere else, Carl Schmitt). The problem is that in these difficult times (on which the special issue for which I have specially written this paper is making focus) those conceptual frameworks are, from my point of view, unable to give us a satisfactory answer. The more I work asking Deleuze these practical questions (as the issue of political actions in this article), the more I find new perspectives to analyze various current social phenomena (specially in Latin America). The way on which Deleuzian ontology is able to bridge the micropolitical and the macropolitical (and the particular way in which he determines the concepts of “micro” and “macro”) is unique and powerful. I will work in the final version in order to make this point clear, and I will make explicit the objective of tackling issues that are truisms in other frameworks (if you allow me, I can mention Arendt and Weber, as you well point our, as examples of this). I will also go further in showing the nature of Deleuzian Ideas as a technical and ontological concept that is developed in Difference and repetition, and I will refer to further reading (both of the Deleuzian literature and myself) for those readers that could be interested in going even beyond. The concept of Idea, along with those of intensity and extension, constitute the Deleuzian ontology, and their form of relation is unique. To show that this does not mean resigning the concepts of action and macro-politics (as many Deleuzians believe) is the contribution that this papers aims to achieve.

As for the model of the artist, I will clarify that the kind of creation of the politician is specific, and should be distinguished from other forms of creation (not only artistic, but also scientific and philosophical); it’s not at all the question to use the artist as a model, but to determine the specificity of the politician through the nature of the ideas he/she should create. I will work hard on this point.

I thank you again for your insights, and I hope you will be satisfied by the final version.

Reviewer 3 Report

This article turns to Deleuze’s under-researched Cinema books, as well as some South American scholarship to address the political problem of political imagination in a despairing world. The author begins by painting a bleak portrait of a contemporary politics and asks the question ‘what is to be done?’, whilst problematising the nature of the Idea as developed within its singular context. How can we kickstart a creative politics, one that brings us out of our current ‘cynical and defeatist’ conditions, given that our imaginary is conditioned by just such an environment.  In answer, the author problematises the ‘double bind of macro- and micro-politics’ in the work of Deleuze, arguing that we must understand these two in an immanent fusion that also takes into account ‘intensities and extensions, flights and individuations’ (135). The politician is then highlighted as the person to mediate the immanent fusion of all these factors and help us generate the proper action that takes us from one situation to the next.

Certainly, not enough ink has been written about the Cinema books and, although they are, in William’s opinion, a poor introduction to the more full-throated metaphysics as found in Difference and Repetition, there are no doubt gems to be yet gleamed in its pages. For attempting to bring these books into relation with Deleuze’s more political philosophy and filling a gap in the literature the author should be commended. A more historiographical account of the notion of Idea in Deleuze’s work, with greater attention to detail, is something the field of study urgently needs. Likewise, this is a burgeoning number of authors in South America writing in the field of Deleuze studies who should be integrated more fully into the secondary literature. That the author explicitly turns to and foregrounds these authors is another string the their bow. 

Nevertheless, I cannot recommend this article for publication for some fundamental reasons:

  1. When engaging with a writer in a discipline with a secondary literature as large as is found in Deleuze studies, a reader would reasonably expect the author to have engaged with this literature to a decent extent. The author does include a number of commentators, though there are claims made within the transcript that have had thousands of pages written about them by authors not found in the bibliography. For example, Patton’s Deleuze and the Political is canonical and would provide the author with much-needed context to inform the relationship between micro- and macro-politics; their conclusion that ‘every politics is simultaneously a macropolitics and a micropolitics’ is drawn from Deleuze himself, and Patton goes to lengths to explain the relationship between the two poles. 
  2. There are too many theoretical discrepancies between the author’s interpretation of Deleuze’s work and those found within not just my understanding of his work, but also key figures in the literature. I have noted some specific examples in the detailed comments below however, in general, I think the author must defend their reading of Deleuze’s tripartite virtual/intensive/actual when they outline it against that in the primary/secondary reading. They must also outline much more carefully and patiently the difference between the argument they are suggesting and Deleuze’s own picture.
  3. However, it is not clear what, other than the above, what substantive contributions the author is making to the literature in this article. There does not seem to be any disagreement between the author and Deleuze’s conclusion. Deleuze would wholeheartedly agree that ‘[i]n order for resistance to become political construction and for flight to take us to habitable spaces, it is necessary for the micro to find its macro dimension’ (693-4). The author even alludes to lines of flight, language that Deleuze uses in A Thousand Plateaus whilst referring to a machinic assemblage that would see to do much of the heavy lifting the author is looking for. So, despite some misgivings about the accuracy of the philosophical reading, they seem to end up at the top of the mountain together. The author should consider foregrounding the thesis statement earlier and much more clearly in the writing.
  4. It is also not clear what the ramifications of the conclusion made might be, and these should be made much clearer. It is slightly strange to see a defence of ‘the politician’ an an essay on Deleuze’s work without any revision; matching the idea of the politician to the integration of micro- and macro-politics does not seem to match the spirit of what Deleuze calls ‘becoming-woman’ and the critique of the War Machine. Is the author pointing towards an ideal politics, or are they simply re-vivifying a particular idea of politics? More specificity here is needed to fully understand the practical import of the metaphysics at work in the paper. Indeed, the notion of politics is used throughout as distinct from ‘micro-’ and macro-political’. Some clarity around idea of what is meant by the first term, and what is meant by ‘the politician’ would add necessary weight to the conclusion. As it stands, it is difficult to see how this article is anything other than an apology for representative politics - and Deleuze is surely not needed for that?

Before developing more specific comments, I will take the liberty of suggesting the author looks at the work of Alain Badiou (Being and Event, Metapolitics, Theory of the Subject), who I think is much more suited to the argument they are making. He develops notions found throughout the transcript (forcing, situation and its relationship to creation) in ways I think will be more conducive to the author’s conclusions.

Specific Comments

  • 45: Why the algebra? What does this contribute? The author aught to bear in mind that their reader will not necessarily have read Deleuze as they are writing this for a non-specialist journal, so dropping this in here is unhelpful.
  • 55-6: It is unclear from this expression whether or not the author things that it is possible for the (‘main’) fields of extension, intensity and Ideas to not articulate immanently. It is also unclear what the authors thinks are the other fields in Deleuze’s work that might constitute non-main (or minor?) fields - and how these interact with the main fields. The secondary authorship on Deleuze’s metaphysics is now so developed that it is reasonable to expect a relatively high degree of specificity in the use of Deleuzian terminology. See Lundy, Smith, Pisters, and Williams as examples of authors who have extrapolated these terms. 
    • This is particularly difficult when the author claims that ‘…it is legitimate to relate the elaborations on cinema with the fundamental ontology developed in the 1968 book’ (49-51). Given that Deleuze argues, in this 1968 book and elsewhere, that relations are external to their terms and that relations precede the terms by which they are defined, it is not clear what is meant by the distinction between legitimate and illegitimate (nor that between mandatory or non-mandatory further down).
  • 101-10: Novel readings of philosophy present important opportunities to challenge dogma and develop new approaches. However, and as noted above, in the reading of a significant figure’s work, if one is to present a reading that flies against a more-or-less established interpretation, it is standard academic practice to reference both the primary literature and at least some of the established secondary literature. One might also hope that an author explains why they argue differently. In this paragraph, the author distinguishes between Deleuze’s political writings and some apparently a-political writing (yet does not specify what constitutes what), whilst making a claim about the nature of Ideas in his work that, as far as I am aware of, flies in the face of a standard interpretation of Deleuze’s work. I do not think that the Idea is limited to what the author implies to be the macropolitical (‘Deleuze restricts Ideas to the great social structures only in the political field’) and I believe that this is a fundamental misinterpretation of Deleuze’s philosophy. Even if the author is correct to argue this, articulating the difference between the political and a-political work might be helpful to the reader. Readers will likely remain open to being convinced by an argument to be made, not assumed. Unfortunately, this is an issue that crops up through the transcript.
  • 215-7: ‘Thus the generalities and the laws of nature appear. Thus individuals and genres appear. Thus the perseverations and the constants appear’. One might think that, if we proceed from the more to less differentiated and productive to a reduced difference, individuals would appear before before generalities and laws of nature. Regardless, it is not clear what the author means here when referring to ‘appearance’. Is this literally ocular, or do they mean metaphysically? If the latter, how does the author reconcile this statement with Deleuze’s focus upon the immanent (see Lundy’s Bergsonism and Ansell-Pearson’s Germinal Life)?
  • 322-26: It is not clear who is suggesting that the extensive field is enough, nor what this might entail. Deleuze is clear throughout D&R that the tripartite virtual/intensive/actual are all significant in his philosophy, and Clisby’s ‘Deleuze's Secret Dualism?’ articulates what is at stake in the various priorities afforded each within the secondary literature’s interpretations.
  • 358-9: It is a mischaracterisation of Deleuze’s metaphysical position to infer that things happen ‘in’ intensity. Deleuze’s correction of Kant’s ‘copernican revolution’ was to show that there are no a priori planes in which things happen (for Kant, events transpired within time and space) - this is exactly what the immanence to which the author refers implies. The author seems to recognise this in the following section, so perhaps this is only an issue with expression, but I am not sure.
    • 583-92: Unless what the author refers to by the use of the term ‘minor artist’ in this passage does not mean what one might commonly understand, they cannot be found in either the virtual (or even the intensive), according to Deleuze. For Deleuze, differentiated identity is only to be found in the actual, though artists themselves will be comprised (as all things are) of all three. The author may not refer to what is commonly understood as a minor artists when they use this term but, if this is the case, what the author is referring to needs to be made more explicit. On the other hand, this might be the author’s own argument and a criticism of Deleuze’s work. If this is the case, the argument is unconvincing given the breadth and depth of literature on the virtual/intensive/actual that has been written both by Deleuze and the secondary literature. Put simply: I do not think that Ideas are virtual relations and my reading is consistent with the broad spectrum of Deleuze readers (thought, again, I remain open to being convinced otherwise). Additionally: are Ideas virtual relations or are they structure-Ideas (646)? I do not see how they can be both, though perhaps the author can explain.
    • As above: the primary and secondary literature has not been substantively engaged with and therefore it is difficult to see either why we should accept the assertion or how the article is to make a significant contribution to the literature on Deleuze. A much closer reading (and one that takes less poetic liberty with detail) of both Deleuze’s own work and that written about him is needed to do either. 
  • 481: Asking a rhetorical question from the literature and offering an answer whilst explicitly brushing off the importance of the technical term in the question is unhelpful. Without knowing the problem that the answer is addressing, the reader is left with a suggestion of some form of action that may or may not be helpful. This should either be addressed in a footnote or removed.
  • 553-4: Why is this famous quote from Fredrick Jameson not attributed to him?

Author Response

First of all, I would like to thank you for the attentive reading of my text, and for highlighting the need of fully integrating the “burgeoning number of authors in South America” into the secondary literature on Deleuze. I also appreciate that you consider of value my effort of bringing the Deleuzian ontology into relation with his political philosophy, to which I have dedicated several decades. I also appreciate your notes, which point towards blind spots in my paper that I hope to cover in the final version of the manuscript.

1. One of the main objections that you rise is the absence of the key figures of literature (Patton, Lundy, Smith, Pisters, Williams, Clisby, and yourself). However, the aim of this paper is to “focus on the South American scholars” (439), in order to contribute to their fully integration into the secondary literature of Deleuze. It is true that I only point that out in section 4, the one on intensity, and I should do the same regarding other concepts, as Idea and extension, where the South American scholars have widely contributed to a different perspective than the literature in English and French. However, it is true that references to some of the authors you mention can be useful, and will allow to be more explicit on the difference between the “standard Deleuzianism” in the central countries and South America. Particularly, the tripartition virtual/intensive/extensive is hegemonic now in our region. The problem is that the aim of this paper is not to develop the ontological debate (which I have lengthily done in other books and papers, both individual and collective), and it is not either the aim of the Special Issue for which it was written. Nonetheless, I strongly believe that I can find a middle ground, referring to the debate in the English-speaking field (I specially appreciate Clisby’s account, that has been translated to Spanish and lengthily discussed) and bringing some of my main arguments and Deleuze’s quotes that support my personal interpretation. I will also include Patton’s position regarding the relation between macro and micro politics. While doing so, I will make more clear with is my contribution to the literature (which according to your reason 3, is not clear enough). I believe that this way I can give place to your observations, while keeping the South American spirit of the paper.

2. Regarding reason nº 4, I will clarify the place of “representative politics” and why I believe that the need of this macro-political tool does not contradict the fundamentals of Deleuze’s position. Of course, it means I new perspective on “representation”, that has nothing to do either with the transcendent ground of the classical political philosophy, nor with the immanence of Capitalism and the rules of the market. I will show that it matches concepts as “becoming” or “War Machine”, as long as we keep in mind the pertinent ontological distinctions, and consider their specific contributions to finding a way out of our current despairing situation.

3. I would like very much some day to discuss in length with you why I believe the Deleuzian frame of work is more useful in the political field than that of Badiou.

4. I thank you for the specific comments, which point towards clarifications and some problems of expression (due mainly to the fact that English is not my native language), and I will work on them. It seems however to be a discrepancy between the line numbers you have and the ones in the version of the manuscript than the Journal made available to me; in most cases this is not an issue, but in other it makes a little hard to place the issue in stake. I don’t know if there is a way to fix this (can you maybe attach your version?), but it would be useful.

I thank you again for your insights, and I hope you will be satisfied by the final version.

Round 2

Reviewer 2 Report

The revised version is much improved, and indeed addresses many of the points I had raised.

Author Response

I'm glad to hear that you find the revised version satisfactory.

Best regards

.

Reviewer 3 Report

This version of the article is a substantial improvement on the first. The author has responded to a number of the comments I made in the feedback, and I thank them for taking them constructively, as they were intended. It is much clearer now that the author takes into account the Anglo/French reading of Deleuze and disagrees with it, preferring instead to foreground a South American reading. Perhaps there is literature out there which substantiates the differences succinctly, as I thought this version could have benefitted from a clearer outline of these differences but, if not, there’s a book proposal in there somewhere. Likewise, more care is taken to consider the actual and the intensive and their role in action than there was in the previous version – these changes are worth the time spent making them.

I remain unconvinced by the argument in this article; however this is not necessarily a problem and I can recommend this article for publication much more than I could previously. That said, I still think there are aspects that could be improved, though I recognize the considerable work done to re-write parts of the article following the previous round.

  1. I must insist that the phrase ‘It is easier to imagine the end of the world than it is to imagine the end of capitalism’ is properly referenced to Fredrick Jameson, who is well-known to have observed this first.
  2. The theme of action, which occupies the first half of the article, is somewhat dropped in the second. In the turn towards the axiological macro/micro duality. I think this could easily be written in, for they are both forms of action and so tying up the loose ends here–relating action to the different politics–could result in the article making a more significant contribution to the literature on Deleuze.
  3. Having made quite the statement about the role of the politician in the previous version, the politician is almost relegated to unimportance in this one. It is not clear why: whilst I was unconvinced by the novelty of the argument, bringing in Deleuze to support representative politics, the author’s feedback led me to believe that this account was going to be more substantive. Given the author’s title, I think this depreciation ought to be reversed.

Author Response

I'm glad to hear that you find the revised version much improved. I look forward to further discussions.

Best regards